# Low-Temperature Vacuum Drying on Broccoli: Enhanced Anti-Inflammatory and Anti-Proliferative Properties Regarding Other Drying Methods

**DOI:** 10.3390/foods12173311

**Published:** 2023-09-02

**Authors:** Antonio Vega-Galvez, Elsa Uribe, Alexis Pasten, Javiera Camus, Michelle Rojas, Vivian Garcia, Michael Araya, Gabriela Valenzuela-Barra, Angara Zambrano, Maria Gabriela Goñi

**Affiliations:** 1Food Engineering Department, Universidad de La Serena, Av. Raúl Bitrán 1305, La Serena 1700000, Chile; muribe@userena.cl (E.U.); afpasten@userena.cl (A.P.); javiera.camus@userena.cl (J.C.); michelle.rojasp@userena.cl (M.R.); vivian.garcia@userena.cl (V.G.); 2Instituto Multidisciplinario de Investigación y Postgrado, Universidad de La Serena, La Serena 1700000, Chile; 3Centro de Investigación y Desarrollo Tecnológico en Algas (CIDTA), Facultad de Ciencias del Mar, Universidad Católica del Norte, Larrondo 1281, Coquimbo 1780000, Chile; mmaraya@ucn.cl; 4Laboratorio de Productos Naturales, Facultad de Ciencias Químicas y Farmacéuticas, Universidad de Chile, Santiago 8380000, Chile; gabadriela@uchile.cl; 5Facultad de Ciencias, Instituto de Bioquímica y Microbiología, Universidad Austral de Chile, Casilla P.O. Box 567, Valdivia 5090000, Chile; angara.zambrano@uach.cl; 6Center for Interdisciplinary Studies on the Nervous System (CISNe), Universidad Austral de Chile, Valdivia 5090000, Chile; 7Grupo de Investigación en Ingeniería en Alimentos, Departamento de Ingeniería Química y Alimentos, Facultad de Ingeniería, Universidad Nacional de Mar del Plata, Mar del Plata 7600, Argentina; ggoni@fi.mdp.edu.ar; 8Consejo Nacional de Investigaciones Científicas y Tecnológicas (CONICET), Buenos Aires 1000-1499, Argentina

**Keywords:** Brassicaceae, bioactive compounds, drying processes, inflammatory inhibitors, MDA-MB-231 cell line

## Abstract

Low-temperature vacuum drying (LTVD) has shown great potential for drying vegetables. It could avoid excessive degradations of active compounds with potential therapeutic agents. In this study, the effect on several relevant bioactive compounds, anti-inflammatory activity, and anti-proliferative activity of broccoli (*Brassica oleracea* var. italica) were evaluated. Effects of other drying methods, including vacuum drying (VD), convective drying (CD), infrared drying (IRD), and freeze drying (FD), were also comparatively evaluated. The results of all dried samples showed high polyunsaturated fatty acid contents (of up to 71.3%) and essential amino acid contents (of up to 8.63%). The LTVD method stands out above the other drying methods, since it obtained the highest content of total phenols, chlorogenic acid, and ferulic acid. Both the LTVD and CD samples demonstrated high anti-inflammatory and anti-proliferative activities. These CD and LTVD samples were also the most active against the breast carcinoma MDA-MB-23 cell line. Due to the good retention of bioactive compounds via LTVD, the obtained dried broccoli here can be used in a near time as an ingredient for the development of novel natural products with anti-inflammatory and anti-proliferative effects.

## 1. Introduction

Broccoli (*Brassica oleracea* L. var. italica) is a popular cash crop and has great economic value. In fact, the market value of the broccoli-related business exceeds one billion dollars [1]. This vegetable is cultivated worldwide, with China and India being the largest global producers, while in North America these are the USA and Mexico, and in Europe these are Spain and Italy [2]. The broccoli popularity amongst consumers is associated with numerous health benefits, as it provides essential nutrients (including vitamins, amino acids, and dietary fiber) and bioactive compounds, such as sulforaphane (SFN), glucosinolate (GLS), and phenolic compounds, among others [3,4]. For instance, epidemiological studies have highlighted the anti-carcinogenic effects of several isothiocyanates from broccoli, specifically SFN, which is a biologically active molecule derived from the enzymatic degradation of glucoraphanin, a specific GLS [5]. The anti-cancer effect of SFN is produced through the modulation of oxidative stress, suppression of metastasis and angiogenesis, and stimulation of apoptosis [6]. In addition, as cancer is linked to inflammatory components, SFN can block the activation of inflammation-related mediators and the gene expression of pro-inflammatory cytokines [7,8,9]. Nonetheless, it is highly unstable, which limits its potential therapeutic applications [10]. As SFN is formed via enzymatic processes, the rates of production and degradation are highly influenced by temperature [11]. Therefore, a minimizing of the SFN-mediated degradation by a regulation of the temperature during the drying process is required.

Recently, a drying method using a low-temperature vacuum (LTVD) has shown to be promising as a mild drying technique. The oxygen-deficient processing environment effectively inhibits the oxidation reaction, prevents color deterioration, and minimizes the loss of biological activity of food materials [12,13,14,15,16]. Its configuration consists of a chamber containing cooled shelves with a cooling unit that ensures a low temperature during this process. In fact, it can be dried at low temperatures since the boiling point of water decreases with a decrease in pressure [17,18]. Therefore, if the operating pressure is decreased in the dryer, it might be able to preserve the contents of SFN and other bioactive compounds as the overall drying time would be reduced [11].

This study aimed to provide a dried broccoli product using low-temperature vacuum drying (LTVD) as an alternative to current methods. LTVD was then compared against standard drying methods related to their impact on nutritional properties, bioactive compounds, and biological activities. The selected standard drying methods include convective drying (CD), infrared drying (IRD), vacuum drying (VD), and freeze-drying (FD). The reasons to select these methods were the following: (i) currently being the most common method employed for the drying of food materials (i.e., CD), (ii) having the best energy efficiency and drying kinetics (i.e., IRD), and (iii) to be between the operating conditions of the LTVD (i.e., VD and FD) based on the literature [19,20]. The acquired knowledge in this study provides a basis for the use of a new approach that minimizes the degradation of bioactive compounds in dried broccoli products and enhance its anti-inflammatory and anti-proliferative effects.

## 2. Materials and Methods

### 2.1. Plant Materials and Drying Methods

Fresh broccoli (*Brassica oleracea* var. italica) was purchased from Dos Marías Company ubicated in Pan de Azúcar, Coquimbo Region, Chile. This cruciferous vegetable was washed and checked carefully to discard spoiled broccoli heads and were then cut into uniform-sized florets. The average initial moisture content of this cruciferous vegetable was 87.25% on wet basis (w.b.). This value was close to that reported in a previous study [21]. The drying process was carried out using different methods, including vacuum drying (VD), convective drying (CD), infrared drying (IRD), low-temperature vacuum drying (LTVD), and freeze-drying (FD). The dried samples reached similar moisture levels, ranging from 8.43% to 9.72% (w.b). Moisture content was determined using the AOAC 934.06 [22] method, and the samples underwent the following drying methods:(a)VD samples were dried using a VO-400 vacuum oven system (Memmert, model VO 400, Schwabach, Germany) connected to a vacuum pump (Büchi, model V-100, Flawil, Switzerland), which controlled, via its solenoid valves, the vacuum inlet. The samples were evenly tiled on a thermoshelf set at 60 °C, and were then vacuum set at 10 kPa for 10.5 h. The end moisture was 9.67% on w.b.(b)CD samples were dried using a designed and constructed convection dryer at the Department of Food Engineering, Universidad de La Serena (Chile). The samples were evenly tiled on a wire mesh tray and convective-dried at 60 °C and a constant air flow rate of 1.5 ± 0.2 m/s. The elapsed time from the beginning of the drying to the end of the drying experiment was 7.5 h, reaching an end moisture of 9.28% on w.b.(c)IRD samples were dried using a designed and constructed infrared drying oven at the Department of Food Engineering, Universidad de La Serena. The samples were evenly tiled on a stainless steel tray and subjected to two 175-Watt IR incandescent lamps mounted inside the oven set at 60 °C for 8.0 h. The distance between the IR source and the samples was 22 cm, with the samples reaching an end moisture of 9.72% on w.b.(d)LTVD samples were dried using a VOcool-400 cooled vacuum oven system (Memmert, model VOcool 400, Schwabach, Germany) equipped with a compact Peltier cooling unit, a thermoshelf (cold and heated), and a vacuum pump. The samples were evenly tiled on the thermoshelf set at 20 °C, and were then vacuum set at 1 kPa for 72 h. The end moisture was 8.71% on w.b.(e)FD samples were dried using an AdVantage Plus vacuum freeze-drying system (Virtis, AdVantage Plus, Gardiner, NY, USA.). The samples were pre-frozen at −80 °C, evenly tiled on a stainless steel tray, and vacuum freeze-dried for 24 h at −60 °C (condenser set point) and 0.027 kPa. The end moisture was 8.43% on w.b.

Once the samples were dried, they were ground into powder using a basic analytical mill (IKA A-11, Wilmington, DE, USA), following which they were sieved, packaged, and stored for further analysis. Each sample was prepared in triplicate.

### 2.2. Fatty Acid Composition

The extraction and methylation of fatty acids in the samples were conducted according to the method of Folch et al. [23]. The conversion of lipids into fatty acid methyl esters (FAMEs) was performed using 14% boron trifluoride-methanol solution (BF_3_-MeOH). Then, FAME extraction was conducted by adding hexane and washing with 20% NaCl. The organic fraction was recovered and evaporated up to dryness and the extract was reconstituted in 1 mL of hexane. The FAMEs were quantified using a gas chromatography (GC) instrument (Clarus 600 FID model, PerkinElmer, USA) equipped with a flame ionization detector (GC-FID) and an Omega Wax 320 capillary column (30 m × 0.320 mm × 0.25 μm, Supelco, St. Louis, MO, USA), having temperature limits of 20–250 °C. The temperature ramp was first maintained at 60 °C for 3 min and then increased by 10 °C min^−1^ to 260 °C at a flow rate of 1.0 mL/min (with nitrogen as a carrier gas). Individual FAs were quantified by comparing the retention times and a peak area of the unknown sample with the FAME standard (Supelco 37 Component FAME Mix, Sigma, nº CRM47885, St. Louis, MO, USA).

### 2.3. Amino Acid Profile

Amino acids were determined using an HPLC pre-column derivatization method [24]. First, 6 N HCl (10 mL) was added to semi-capped hydrolysis tubes containing the sample (200 mg), which were incubated for 24 h at 120 °C in an oven. Then, the solution was transferred to a volumetric flask with a capacity of 50.0 mL and adjusted with distilled water. A 100 μL aliquot of the solution was extracted and adjusted to pH 10.0 within a 1.0 mL borate buffer and was concentrated to dryness in a rotary evaporator. The concentrated extracts were reconstituted to a volume of 200 μL with borate buffer (pH 10.0) and filtered using a 0.22 μm Nylon syringe filter for its derivatization. The amino acids were derivatized with o-phthalaldehyde (OPA) prior to injection and then identified using a ZORBAX Eclipse AAA amino analysis column (3.5 μm, 4.6 × 150 mm). The same mobile phase and chromatographic conditions described by Araya et al. [24] were used.

### 2.4. Bioactive Compound Analysis

#### 2.4.1. Extraction of Bioactive Compounds

Bioactive compound extraction was carried out by weighing 2 g and 1 g of fresh and dried broccoli, respectively. Three extractions were made by adding 10 mL of aqueous MeOH mixture (80%) in the same manner as in our previous study [25]. Finally, the aqueous residue was frozen at −80 °C and then vacuum freeze-dried for 24 h. The freeze-dried extracts were reconstituted in 5 mL of a methanol–formic acid solution (99:1) to obtain a defined volume.

#### 2.4.2. Total Phenolic Content (TPC) and Total Flavonoid Content (TFC)

The TPC and TFC were estimated using spectroscopic methods that employed the Folin–Ciocalteu reagent and aluminum chloride as their complexing agents, respectively. The details of each methodology can be found in a previously published study [14].

#### 2.4.3. Hydroxycinnamic Acids (HCs)

The HCs (chlorogenic acid, ferulic acid, and caffeic acid) were determined using an Agilent Technologies 1200 HPLC system (Waldbronn, Germany) with a C18e (100 × 4.6 mm ID) Chromolith^®^ HighResolution column (Merck, Darmstadt, Germany) and a Diode Array Detector (DAD). The same two mobile phases and chromatographic conditions described by Zúñiga-López et al. [26] were used. A total of 20,000 mg/L of broccoli extract was administered using a 20 µL injection volume. The chromatograms were registered at 280 nm and 322 nm. All HCs were identified by contrasting the retention times and absorbance spectra with the appropriate standard. Quantification was performed using an external calibration curve (caffeic acid: 0.1–2 mg/L [limit of detection (LOD) 0.033 mg/L and limit of quantification (LOQ) 0.111 mg/L]; ferulic acid: 0.1–2 mg/L [LOD 0.045 mg/L and LOQ 0.151 mg/L]; and chlorogenic acid: 0.1–2 mg/L [LOD 0.015 mg/L and LOQ 0.052 mg/L]).

#### 2.4.4. Total Glucosinolate Content (TGC)

The TGC was analyzed according to the method of Aghajanzadeh et al. [27]. Briefly, 60 μL of extracts were mixed with 1800 μL of sodium tetrachloropalladate II (Na_2_PdCl_4_) reagent (2 mM). The reaction solution was then incubated at room temperature for 30 min in the dark before the absorbance was read at 450 nm. The TGC was measured using Sinigrin (Sigma-Aldrich, St. Louis, MO, S1647) as a standard and the results were given as μmol of sinigrin equivalent (SE)/g (d.m.).

#### 2.4.5. Sulforaphane Content (SFN)

The SFN content was determined using an HPLC system. The sample (1 g) was extracted two times with a combination of 20 mL of methylene chloride with 0.5 g of anhydrous ammonium sulfate and agitated for 60 min each time. Then, the solutions were filtered on Whatman#1 filter paper. The combined filtrate was evaporated on a rotary evaporator (BUCHI RE12, Flawil, Switzerland) at 30 °C, according to the procedure described by Gonzalez et al. [28] with some modifications. SFN analysis was performed using an HPLC-DAD (JASCO LC-4000 Series, MD, USA) system. LC separation was carried out on an Epic™ C18 column (5 μm particle size, 250 × 4.6 mm) (PerkinElmer, Waltham, MA, USA). The flow rate was set at 1 mL/min and the mobile phases were set as (A) HPLC-grade water and (B) acetonitrile (100%). The gradient elution program was 0–2 min, 40% A and 60% B; 2–6 min, 30% A and 70% B; 6–8 min, 30% A and 70% B; 8–10 min, 40% A and 60% B; and 15 min, stop. The column oven temperature was set at 30 °C and the absorbance at 254 nm was recorded. Quantification was made through comparison with a sulforaphane standard curve.

### 2.5. Biological Potential Analysis

#### 2.5.1. Anti-Inflammatory Potential

The evaluation of the anti-inflammatory potential consisted of the topical administration of the extracts (3.0 mg/ear), immediately after the inflammatory agent, arachidonic acid (AA) or 12-O-tetradecanoylphorbol-13 (TPA) acetate, was applied to the right ear of each animal. After 1 h after applying AA (2.0 mg/ear) and 6 h after applying TPA (0.5 mg/ear), the animals were euthanized. A 6 mm piece was then removed from both ears with a mechanical punch. Both pieces were weighed and the difference in the weight determined of the edema weight in each animal was measured. The effect was determined based on the median edema weight of control animals (*n* = 16) and treated animals (*n* = 8). The reference drugs used were nimesulide (1.0 mg/ear) and indomethacin (0.5 mg/ear) for AA and TPA, respectively [29].

#### 2.5.2. Anti-Proliferative Potential

##### Cell Culture

The human breast adenocarcinoma MDA-MB-231 cell line (ATCC no. HTB-26) was cultured in RPMI-1640 medium containing 10% fetal bovine serum (FBS), 50 mg/mL streptomycin, 50 U/mL penicillin, 1 mM sodium pyruvate, 2 mM L-glutamine, and non-essential amino acids at 37 °C in a 5% humidified CO_2_ atmosphere. Chemicals were obtained from Hyclone, Logan, UT, USA.

##### Cell Viability Assays

In a 96-well black plate, approximately 5000 MDA-MB-231 cells were seeded in each well. The cells were incubated for 18–24 h and treated with different concentrations of the broccoli extracts that had previously been resuspended in the same medium. After 48 h of treatment, the cells were then treated with 100 µL of HBSS-Ca^2+^ solution (40 µM KH_2_PO_4_; 30 µM NaH_2_PO_4_ x H_2_O; 13.6 µM NaCl; 600 µM D-Glucose; 500 µM KCl; and 900 µM CaCl_2_), which contained 5 µM propidium iodide (PI) as a final concentration. The cells were incubated for an additional 10 min. An automated microtiter plate reader (Metertech S960) with 535 nm excitation and 617 nm emission was used to detect fluorescence. Data are shown as mean ± SEM.

### 2.6. Statistical Analysis

All determinations were conducted in triplicate. The one-way ANOVA was used to analyze the difference between the means of more than two groups, following which the post hoc Duncan’s multiple range test was performed to measure the specific differences between pairs of means using the statistical software Statgraphics Centurion Version 18.1.12 (Statgraphics Technologies, Inc., The Plains, VA, USA) with a statistical significance level of *p* < 0.05. For the anti-inflammatory assay, the Kruskal–Wallis non-parametric test was utilized, and the Mann–Whitney U test was used for individual comparisons.

## 3. Results and Discussion

### 3.1. Fatty Acids

Although lipids are minor constituents in broccoli florets, oxidation of their fatty acids play an important role on the deterioration degree of broccoli. Eighteen fatty acids were identified across the distinctly dried broccoli samples, but nearly all of them were present in trace quantities and only about seven fatty acids were found at the 1% level or higher (Table 1). As reported in the literature, dried broccoli contains significant amounts of polyunsaturated fatty acids (PUFAs), mainly α-linolenic acid and linoleic acid, and the saturated fatty acid (SFA) palmitic acid can also be found [4,30,31]. These detected fatty acids provide a wide variety of bioactivities and health advantages [32]. However, it is worthy to note that the important amount of PUFAs, especially α-linolenic and linoleic acid, make broccoli susceptible to oxidation during storage due to their multiple double bonds [33]. As shown in Table 1, the fatty acid contents of broccoli varied slightly depending on the drying method used. For instance, the broccoli dried under both VD and LTVD had significantly more α-linolenic acid (55.33 g/100 g and 54.91 g/100 g, respectively) than the other dried samples (*p* < 0.05). A common feature in these techniques is the use of a vacuum that provides an oxygen-deficient process environment, thereby diminishing the rate of lipid auto-oxidation. This could explain that a higher content of PUFAs was observed in the dried broccoli under vacuum-mediated conditions than in those not subjected to a vacuum. In contrast, reductions in the levels of several unsaturated fatty acids after drying through CD and IRD may be due to the oxygen-, heat- and light-induced oxidation of lipids, which could lead to an increase in the saturated fractions of the FAMEs [34]. This is a result of the higher amount of palmitic acid in such samples (19.79 g/100 g and 16.00 g/100 g, respectively) and tricosanoic acid (12.70 g/100 g) in the IRD sample.

Since FD involves a heat process at very low temperatures, and there is a low exposure to oxygen, it was expected that FD-broccoli would have a lower amount of SFAs and more PUFAs than CD- and IRD-broccoli. However, these results were not in line with the expectations. The FD samples showed statistically negligible differences regarding their SFA and PUFA contents when compared with the IRD samples (*p* > 0.05) and were found to be similar to the CD samples. It has been hypothesized that FD results in the formation of a porous product, causing lipids to be more accessible to oxygen, increasing the rate of autoxidation reactions [35].

In a general context, in all the samples, the high level of found α-linolenic acid is of high importance, as it is an essential nutrient that reduces cardiovascular disease risk, due to its favorable involvement in vascular inflammation and endothelial dysfunction [36]. Nevertheless, precautions during the storage of dried broccoli must be taken to avoid the oxidation of this unsaturated fatty acid (α-linolenic acid).

### 3.2. Amino Acids

It is important to have information about the free amino acid composition of broccoli, as they are the precursors of secondary plant metabolites, such as glucosinolates [37]. However, amino acids have been reported to be susceptible to the drying conditions; therefore, they could be lost, changed, or destroyed due to their oxidation [38,39]. The amino acid composition of broccoli following the different drying methods is shown in Table 2. There were 13 amino acids that were identified and quantified, and these amino acids were found in all dried samples, but the amounts of most individual amino acids differed slightly depending on the drying method that was employed. Dried broccoli via the LTVD technique possessed slightly higher values of glutamic acid, serine, arginine, isoleucine, and lysine, being such samples that showed a slightly higher percentage (21.42%) of the total accumulated amino acid content compared with the other samples (Table 2).

In all dried samples, glutamic acid was the major amino acid identified followed by aspartic acid. These results are consistent with that reported by Gomes and Rosa [40], who reported glutamic acid and aspartic acid as the most dominant amino acids in eleven broccoli cultivars. Glutamic acid is a known precursor of glutamine, an important nutrient utilized by intestinal cells to maintain their regulation of enterocyte proliferation, modulate inflammatory pathways, such as the nuclear factor κB (NF-κB) and signal transducer and activator of transcription (STAT) signaling pathways and protect against apoptosis and cellular stresses [37,41].

A number of detected essential amino acids in the dried samples, including isoleucine, leucine, lysine, phenylalanine, and valine, typically contribute between 6.37–8.63% of the total free amino acids in a diet. Amongst them, IRD-broccoli had the highest content (8.63 g/100 g d.m.) of essential amino acid accumulation, whereas FD samples presented a significantly lower value (6.37 g/100 g d.m.). The heat generated during drying can produce a thermal hydrolysis of the complex protein molecules, resulting in the release of certain amino acids, whereas others may be involved as a substrate for the Maillard reaction [42]. On the other hand, it is noteworthy that amino acids are also involved in the biosynthesis of glucosinolates. The glucosinolates that are originated from methionine, leucine, alanine, isoleucine, or valine are called aliphatic glucosinolates, whereas those that are derived from tryptophan are indole glucosinolates, and those derived from tyrosine or phenylalanine are termed aromatic glucosinolates [43]. Nonetheless, our results suggest that there is no absolute correlation between the changes in the abundance of the corresponding amino acids after drying and total glucosinolate contents. This clear non-correspondence can be attributed to cell disruption during broccoli batch preparation (chopping or grinding) before drying, since myrosinase can get into contact with the glucosinolates and are transformed into a variety of degradation products, including nitriles, isothiocyanates, epithionitriles, thiocyanates, indoles, oxazolidine-2-thiones, and hydroxynitriles [44].

### 3.3. Changes in the Bioactive Compound Properties

#### 3.3.1. Total Phenolic Content (TPC), Total Flavonoid Content (TFC), and Hydroxycinnamic Acids (HCs)

Phenolic compounds are important bioactive compounds in broccoli [4]. The TPC and TFC in broccoli processed with the five drying methods are shown in Table 3. Higher values of the TPC were found in the samples from LTVD, CD and FD, and no significant difference was determined among them (*p* > 0.05). Since both LTVD and FD operate under a low temperature and vacuum environment, the TPCs were less affected by the heating and oxygen. Meanwhile, the higher temperatures applied during CD may have also reduced TPC degradation due to the shorter exposure time of the sample to the adverse effects of temperature, light, and oxygen [45,46]. These results are in accordance with Vargas et al. [47], who studied the effects of FD, CD, and refractance window drying (RWD) on broccoli, and their results showed no significant differences in the TPC among the studied drying methods.

In the current study, it was observed that the highest TFC values were found in broccoli subjected to FD (7.13 mg QE/g d.m.). It has been shown that the ice crystal sublimation stage during the FD process might trigger a disruption of the plant’s cell structure, resulting in a solvent penetration and hence a better extraction [48]. Although the second-highest value (4.72 mg QE/g d.m.) was found in CD, it was not significantly greater than that found in IRD (4.56 mg QE/g d.m.). This behavior could be explained with the findings of other researchers, who suggested that both heat and far infrared rays might be able to break down covalent bonds and release certain flavonoids, most of which are located in a bound form in the plant matrix [47,49]. In contrast, the lowest TFC values were found in the LTVD and VD samples (Table 3). As we all know, the vacuum drying’s convective heat transfer decreases the boiling point of the dried materials [50]. We speculated whether it would be able to generate a proper environment for many key enzymes’ activity on flavonoid metabolism and being able to promote the accumulation of specific compounds, such as flavanols, dihydroflavones, chalcones, dihydroflavonols, flavonoid carbonosides, among others [51], that failed to be detected using the spectrophotometric techniques. However, more studies are required to support this speculation.

The HCs are phenolic compounds that act as powerful antioxidants, and three dominant HCs from dried broccoli have been analyzed to reveal the effects of these drying methods. As listed in Table 3, the value of individuals HCs was significantly varied with the different methods. Dried broccoli via the LTVD method had higher contents of chlorogenic acid and ferulic acid, whereas broccoli dried through the FD method demonstrated higher contents of caffeic acid. These results were expected, as HCs are heat-sensitive and both lower heat and oxygen-deficient conditions can prevent an excessive degradation of them [52]. Consequently, a severe level of degradation was found in these phenolic compounds in the dried samples under thermal drying (VD, CD, and IRD), potentially due to oxidation [50]. In fact, it has been previously reported that the *O*-diphenolic hydroxyl groups of the chlorogenic acid molecules were susceptible to oxidative decomposition at high temperatures [53]. Regarding caffeic acid, it was obviously affected by the drying temperature, since it was not detected in broccoli dried via the VD, CD, and IRD methods (Table 3). Only ferulic acid in the CD sample was not majorly affected by the heat, since the amount formed was similar to that of the FD sample (*p* > 0.05). Similar findings were reported by Vargas et al. [47], who showed no significant differences in their dried broccoli via the CD and FD methods as such. This may be due to the hot air used in the CD method that encourages the surface hardening of the sample, thereby preventing the excessive degradation of ferulic acid and other compounds during this process [54].

#### 3.3.2. Total Glucosinolate Content (TGC) and Sulforaphane Content (SFN)

Glucosinolates (GLSs) represent a specialized class of sulfur- and nitrogen-containing metabolites almost exclusive in Brassicaceae that can be hydrolyzed (under the action of myrosinase) to form isothiocyanates (such as SFN), known to impart high levels of biological activity [55]. Due to this reason, the influence of different drying methods on the total glucosinolate content (TGC) and sulforaphane content (SFN) were evaluated in dried broccoli (Table 3). Results showed that the TGC was significantly higher in the FD samples (39.75 μmol SE/g d.m.), followed by the CD samples (32.49 μmol SE/g d.m.), although this last value was only significantly higher than its VD counterpart (Table 3). A number of authors have reported that the TGC in broccoli is better retained upon being dried via the FD method than with the other drying methods [44,47,56]. This could be attributed to the preparation of raw broccoli for freeze-drying by first subjecting it to freezing at −80 °C, which could terminate myrosinase activity, thereby reducing the hydrolysis of the GLSs by myrosinase [57]. This assertion is inferred from the fact that the SFN contents (2.43 mg/g d.m.) in the FD samples were significantly lower than the broccoli samples dried using other methods (Table 3), which implies that there was a minimized transformation of glucoraphanin into SFN in such samples.

Controlling the temperature is one of the primary requirements to minimize losses in the GLSs during processing. LTVD operates between traditional VD and FD [19], thus it also employed a low temperature and a strong vacuum for dehydration, but in absence of a freezing step followed by sublimation [13,17]. However, the low temperature employed during LTVD (20 °C) was not low enough to stop the myrosinase action. Thus, its activity may have continued during the drying process to a certain extent, which provides a strong basis for the significantly lower TGC and higher SFN detected in the LTVD samples compared to its FD counterpart.

In this study, the temperature of the VD, CD, and IRD methods was set at 60 °C; this is a temperature at which myrosinase still remained active [46,57,58,59]. Therefore, it can be expected that variations in these drying processes and conditions (radiation energy, drying time, air velocity, vacuum, etc.) could cause different responses in the myrosinase–GLS interaction mechanism for broccoli [47,55]. Comparatively, the VD method formed the lowest TGC among the CD and IRD methods. It could be speculated that low oxygen concentrations and bubble formation due to vacuum conditions might have predisposed the material to such stress factors that disrupted the cells, thereby facilitating the myrosinase–GLS interaction to produce SFN. The CD method retained more TGC than their VD and IRD counterparts, apparently due to the fact that the water loss was faster than the aforementioned methods, and thereby retarding the myrosinase activity. It is also believed that water loss increases the cell membrane’s permeability and facilitates contact between the myrosinase and protease enzymes, indirectly weakening its capacity to degrade GLSs [60]. Surprisingly, the SFN content was observed to be higher in the CD samples. This behavior was contrary to what was expected. We speculated that the conditions employed during the CD process was not enough to inactivate the myrosinase; thus, its activity might have continued during the storage procedure until SFN extraction and quantification was conducted. However, the reason behind this phenomenon still needs to be clarified. Some authors have confirmed that the residual myrosinase after drying and even post blanching could cause a sufficient enzymatic degradation of the GLSs during the storage process [44,57]. Although the drying time in the CD method is resembled to that of the IRD method, the content of SFN was diminished to a greater extent during IRD. The infrared energy absorbed by the food material in different layers generates vibrating movements of the water molecules and causes them to fluctuate to produce heat [20]. These fluctuations probably facilitate the myrosinase–GLS interaction mechanism to enhancement enzymatic activity. However, the food products could overheat during IRD due to different parameters (intensity, infrared power, wavelength, and distance), promoting the thermal degradation of SFN.

Overall, it could be said that the TGC and SFN content are highly dependent on the drying method used [47,55]. However, more studies are needed to explore the combined effects of myrosinase activity, degradation of the GLSs (mainly glucoraphanin), and generation of SFN during various drying processes to further understand these mechanisms. On the other hand, a dozen factors should be considered that affect the formation and degradation of SFN, embracing the activities of myrosinase, temperature, water content and the epithio-specifier protein, contents of vitamin C, Fe^2+^, phenolics, etc. [60], which further complicates the understanding of the pattern of SFN variation in broccoli under drying.

### 3.4. Changes in the Biological Potential Properties

#### 3.4.1. Anti-Inflammatory Potential

To evaluate the anti-inflammatory potential of broccoli extracts, the AA and TPA-induced inflammation assays were chosen as models for acute and chronic inflammation, respectively. On the one hand, the topical application of AA triggers a rapid inflammatory process linked to elevated levels of inflammatory mediators, like prostaglandin E_2_ (PGE_2_), leukotriene B_4_ (LTB_4_), and thromboxane A_2_ (TXA_2_), and other oxidized derivatives generated from AA metabolism through distinct enzymes, i.e., cyclooxygenase (COX) and lipoxygenases (LOXs), which result in an intense edema, erythema, and extravasation of the leukocytes and proteins [61,62]. While, on the other hand, the topical administration of TPA causes a pleiotropic tissue reaction that includes a significant inflammatory response brought on by the over-expression of inflammatory mediators, such as the interleukins (IL)-6 and IL-1β, the tumor necrosis factor-α (TNF-α), the inducible nitric oxide synthase (iNOS), and the cyclooxygenase-2 (COX-2), that cause oxidative stress, epidermal hyperplasia, neutrophil and macrophage infiltration, T cell proliferation, and edema [61,63,64].

When each phlogistic agent was applied to the right ear of the respective test group mice, it generated significant inflammation and increased epidermal thickness regarding the left ear. In most cases, the application of different drying methods to broccoli demonstrated a significant anti-inflammatory effect of their extracts (Table 4). Only the VD-broccoli extract did not significantly inhibit edema formation in either model used. In contrast, the extracts obtained from broccoli dried via LTVD method exhibited a considerable inhibition of edema in both inflammation models (Table 4). In fact, the edema induced by AA was inhibited more effectively by such extracts than even for nimesulide (55.4% and 53.4%, respectively), a non-steroidal anti-inflammatory drug (NSAID) which was utilized as a positive control in our work. These findings suggest that the anti-inflammatory effect of the broccoli extracts obtained through LTVD may contain compounds that are able to limit the development of edema and inflammation by modulating certain molecular processes related to the synthesis of lipid mediators [61,65]. In fact, such extract has the highest contents of chlorogenic and ferulic acids (see Table 3) that protects against oxidative stress and, as a result, reduces pro-inflammation [66,67]. Although the FD samples revealed higher chlorogenic acid and caffeic acid values than the samples dried by CD, they showed lower levels of anti-inflammatory activity. This low activity was probably caused by the low content of SFN in the FD samples. In fact, the second-most active anti-inflammatory extract was obtained from broccoli dried by CD, probably because it kept the highest contents of SFN (see Table 3). There is evidence that broccoli-extracted compounds, such as HCs (chlorogenic, caffeic, and ferulic acids) and SFN, decrease the levels of several inflammatory mediators. For example, chlorogenic acid included in the cruciferous vegetables could only block the production of IL-6 at high levels (25 µM and 50 µM), whereas SFN could perform the same at low doses (10 µM), while simultaneously inhibiting the production of IL-1β, IL-6, and TNF-α [68]. Other studies showed the same trend, reporting that SFN from broccoli caused an inhibition of IL-6 in human cancer HepG2 cell lines [6], in addition to inhibiting TNF-α, IL-1β, and IL-6 in RAW 264.7 cells [7]. Inhibition of COX-2 and LOX enzymatic activities in the extracts of broccoli sprouts after an elevated CO_2_ treatment was also revealed [37].

According to the results presented in this study, it is deduced that the analyzed compounds might act individually as anti-inflammatory agents or in a synergistic form to block inflammation-related mediators. However, it will depend on the complexity of the food matrix after drying and how the used method affects these compounds.

#### 3.4.2. Anti-Proliferative Potential

The effect of broccoli extracts on the viability of MDA-MB-231 breast adenocarcinoma cells at concentrations 0.3–2.5 mg/mL is shown in Figure 1. A significant decrease in the number of viable cells compared to the negative control group can be noted after treatment with different concentrations. At 24 h of exposure, the extracts obtained from the dried samples by CD and LTVD were the most active against MDA-MB-231, with inhibitory rates of 63.89% and 60.69% at 0.6 mg/mL of concentration, respectively. In fact, such extracts with a concentration of 1.2 mg/mL were able to reduce 100% of cancer cell viability, while the inhibitory rate of the positive control dimethylformamide (DMF) only reached 90.46% (Figure 1).

In order to obtain a standard indicator for rating the efficacy of broccoli extracts in inhibiting breast adenocarcinoma cells, the IC50 (half maximal inhibitory concentration) was computed (Table 5). The IC50 values for the CD extract and LTVD extract were calculated to be 0.396 mg/mL and 0.433 mg/mL, respectively. In the literature, the IC50 value for the extracts from dried broccoli by hot-air-drying (0.570 mg/mL) was similar to the value identified in the current study [3]. The authors demonstrated that such an extract exhibited the highest cytotoxic effect on MDA-MB-231 cells in relation to extracts obtained from cooked broccoli through steaming or a microwave prior to hot air drying. As part of our results, it is clear that the extracts obtained from dried broccoli through the CD and LTVD methods induced a higher inhibitory effect on the MDA-MB-231 cell line compared to the other methods. These results can be attributed, among other compounds, to the abundant presence of chlorogenic acid and ferulic acid in the LTVD samples, which are compounds that are considered to have anti-cancer potential due to its ability to induce apoptosis and autophagic cell death depending on intracellular reactive oxygen species production in numerous cancer cell lines [69,70]. SFN also exhibits cytotoxicity against cancer cell lines, with this compound being the most abundant in the CD samples. Some authors have proposed that the mechanism of action of SFN against MDA-MB-231 breast cancer cells has been related to its effect to induce apoptosis [71,72,73,74,75,76,77], expression of pro-apoptotic genes [73,74], epigenetic regulation [76,78,79], and G2/M cell cycle arrest [71,72,75,76,79]. This study presents preliminary results that will be beneficial for the evaluation of the bioactive potential of dried broccoli samples and the development of novel chemotherapeutic agents. However, further studies will be needed to investigate the precise mechanisms on the cytotoxic activities of these extracts against cancer cell lines.

## 4. Conclusions

The current study responds to expectations about minimizing the degradation of highly labile bioactive compounds in broccoli subjected to different drying methods. The LTVD method showed the best results in the retention of heat-sensitive phenolic compounds (chlorogenic and ferulic acids), whereas CD was helpful to enhance SFN content. In comparison, LTVD extracts performed better regarding anti-inflammatory effects on AA-induced ear edema, while CD extracts were somewhat more active against MDA-MB-231 breast adenocarcinoma cells than LTVD extracts. Surprisingly, FD resulted in lower anti-inflammatory and anti-proliferative activities than LTVD and CD. Probably, the freeze conditions previously applied to FD could stop myrosinase activity, preventing the formation of SFN. Due to the good functional properties and bioactive components of broccoli, both LTVD and CD (in this order) proved to be the most suitable drying methods for producing dried products with anti-inflammatory and anti-proliferative protective agents. However, the large drying time and energy consumption in LTVD are limitations that cannot be ignored, so combination with adequate pre-treatment techniques to reduce the drying time still needs further research in the future.

## Figures and Tables

**Figure 1 foods-12-03311-f001:**
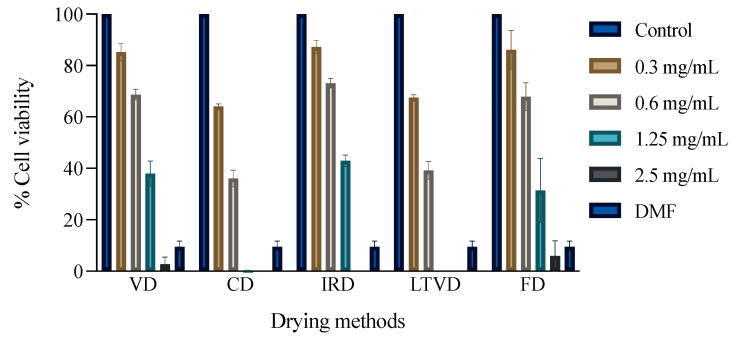
Anti-proliferative effects estimated through the propidium iodide (PI) assay of extracts from dried broccoli using different methods against MDA-MB-231 human cancer cell lines. Data are expressed as mean ± SEM of three independent experiments (*n* = 3). Dimethylformamide (DMF) was used as a positive control. Negative control: 100% cell viability.

**Table 1 foods-12-03311-t001:** Effect of different drying methods on fatty acid composition of broccoli.

Fatty Acid (g/100 g)	Drying Methods				
	VD	CD	IRD	LTVD	FD
Saturated Fatty Acids					
Lauric acid, C12:0	0.24 ± 0.03 ^b^	0.66 ± 0.10 ^a^	0.15 ± 0.01 ^b^	0.23 ± 0.02 ^b^	0.17 ± 0.01 ^b^
Myristic acid, C14:0	0.48 ± 0.04 ^a^	0.24 ± 0.04 ^d^	0.41 ± 0.04 ^ab^	0.36 ± 0.02 ^bc^	0.33 ± 0.02 ^c^
Pentadecanoic acid, C15:0	0.19 ± 0.02 ^ab^	0.17 ± 0.02 ^ab^	0.20 ± 0.01 ^a^	0.17 ± 0.02 ^ab^	0.16 ± 0.01 ^b^
Palmitic acid, C16:0	14.82 ± 0.33 ^bc^	19.79 ± 1.03 ^a^	16.00 ± 0.81 ^b^	14.66 ± 0.40 ^bc^	14.03 ± 0.52 ^c^
Heptadecanoic acid, C17:0	0.17 ± 0.01 ^b^	0.15 ± 0.01 ^b^	0.16 ± 0.01 ^b^	0.15 ± 0.00 ^b^	3.15 ± 0.13 ^a^
Stearic acid, C18:0	1.42 ± 0.03 ^ab^	1.90 ± 0.49 ^a^	1.36 ± 0.02 ^b^	1.28 ± 0.01 ^b^	1.21 ± 0.02 ^b^
Arachidic acid, C20:0	0.31 ± 0.04 ^a^	0.33 ± 0.13 ^a^	0.27 ± 0.04 ^a^	0.25 ± 0.01 ^a^	0.34 ± 0.00 ^a^
Behenic acid, C22:0	0.11 ± 0.03 ^bc^	0.61 ± 0.07 ^a^	0.11 ± 0.06 ^bc^	0.02 ± 0.01 ^c^	0.15 ± 0.01 ^b^
Tricosanoic acid, C23:0	4.59 ± 0.41 ^bc^	3.70 ± 0.40 ^c^	12.70 ± 1.71 ^a^	7.57 ± 0.16 ^b^	11.21 ± 3.07 ^a^
Lignoceric acid, C24:0	0.27 ± 0.03 ^b^	0.28 ± 0.06 ^b^	0.26 ± 0.06 ^b^	0.27 ± 0.02 ^b^	0.43 ± 0.05 ^a^
Monounsaturated fatty acids					
Palmitoleic acid, C16:1-*cis* (n-7)	2.67 ± 0.01 ^a^	2.08 ± 0.06 ^c^	2.00 ± 0.06 ^c^	1.83 ± 0.02 ^d^	2.29 ± 0.06 ^b^
Heptadecenoic acid, C17:1 (*cis*-10)	2.86 ± 0.10 ^a^	1.84 ± 0.49 ^b^	2.43 ± 0.15 ^a^	2.50 ± 0.10 ^a^	0.18 ± 0.00 ^c^
Oleic acid/trans-oleic acid, C18:1-*cis* (n-9)/C18:1-*trans* (n-9)	0.33 ± 0.01 ^b^	0.52 ± 0.08 ^a^	0.56 ± 0.01 ^a^	0.34 ± 0.01 ^b^	0.29 ± 0.02 ^b^
11-Eicosenoic acid, C20:1-*cis* (n-9)	ND	ND	ND	ND	0.04 ± 0.00 ^a^
Nervonic acid, C24:1-*cis* (n-9)	0.24 ± 0.03 ^ab^	ND	0.22 ± 0.02 ^b^	0.25 ± 0.03 ^ab^	0.28 ± 0.00 ^a^
Polyunsaturated Fatty Acids					
Linoleic acid, C18:2-*cis* (n-6)	15.50 ± 0.16 ^b^	16.34 ± 0.10 ^a^	13.08 ± 0.30 ^e^	14.72 ± 0.11 ^c^	13.83 ± 0.43 ^d^
α-linolenic acid, C18:3-cis (n-3)	55.33 ± 0.51 ^a^	51.35 ± 0.47 ^b^	49.58 ± 0.66 ^b^	54.91 ± 0.36 ^a^	51.09 ± 1.79 ^b^
γ-linolenic acid, C18:3-cis (n-6)	0.48 ± 0.02 ^b^	ND	0.50 ± 0.01 ^b^	0.49 ± 0.00 ^b^	0.55 ± 0.02 ^b^
SFA ^1^	22.59 ± 0.55 ^c^	27.84 ± 1.02 ^b^	31.62 ± 1.03 ^a^	24.96 ± 0.37 ^c^	31.18 ± 2.33 ^a^
MUFA ^2^	6.10 ± 0.11 ^a^	4.44 ± 0.61 ^c^	5.21 ± 0.07 ^b^	4.92 ± 0.10 ^bc^	3.09 ± 0.08 ^d^
PUFA ^3^	71.31 ± 0.66 ^a^	67.77 ± 0.56 ^bc^	63.16 ± 0.95 ^d^	70.12 ± 0.47 ^ab^	65.46 ± 2.23 ^cd^

Data are expressed as mean ± standard deviation (*n* = 3). Significant differences (*p* < 0.05) are indicated with different superscript letters in the same row. ND: not detected at a level <0.01 g/100 g sample. ^1^ Saturated fatty acids; ^2^ monounsaturated fatty acids; ^3^ polyunsaturated fatty acids.

**Table 2 foods-12-03311-t002:** Effect of the drying method used on the amino acid profile of broccoli.

Amino Acids (g/100 g d.m.)	Drying Method				
	VD	CD	IRD	LTVD	FD
Aspartic acid	2.10 ± 0.39 ^a^	1.72 ± 0.15 ^a^	2.08 ± 0.16 ^a^	1.81 ± 0.31 ^a^	2.13 ± 0.04 ^a^
Glutamic acid	3.34 ± 0.59 ^ab^	2.69 ± 0.23 ^b^	3.29 ± 0.47 ^ab^	3.95 ± 0.54 ^a^	3.86 ± 0.30 ^a^
Serine	1.09 ± 0.20 ^b^	1.06 ± 0.08 ^b^	1.31 ± 0.11 ^ab^	1.41 ± 0.16 ^a^	1.21 ± 0.02 ^a^
Glycine	0.85 ± 0.13 ^c^	1.07 ± 0.06 ^b^	1.32 ± 0.10 ^a^	1.28 ± 0.10 ^a^	0.42 ± 0.03 ^d^
Threonine *	1.15 ± 0.21 ^ab^	0.87 ± 0.14 ^bc^	1.19 ± 0.15 ^a^	1.03 ± 0.03 ^ab^	0.73 ± 0.03 ^c^
Arginine	1.29 ± 0.26 ^bc^	1.10 ± 0.05 ^c^	1.55 ± 0.14 ^b^	2.59 ± 0.16 ^a^	2.47 ± 0.01 ^a^
Alanine	1.53 ± 0.28 ^a^	1.63 ± 0.13 ^a^	1.94 ± 0.15 ^a^	1.65 ± 0.25 ^a^	0.89 ± 0.10 ^b^
Tyrosine	0.50 ± 0.04 ^b^	0.50 ± 0.02 ^b^	0.64 ± 0.08 ^a^	0.53 ± 0.06 ^b^	0.48 ± 0.01 ^b^
Valine *	0.99 ± 0.19 ^ab^	0.88 ± 0.09 ^b^	1.15 ± 0.06 ^a^	0.94 ± 0.16 ^ab^	0.98 ± 0.01 ^ab^
Phenylalanine *	1.55 ± 0.28 ^ab^	1.37 ± 0.12 ^ab^	1.62 ± 0.00 ^a^	1.43 ± 0.32 ^ab^	1.17 ± 0.11 ^b^
Isoleucine *	1.89 ± 0.29 ^bc^	2.02 ± 0.19 ^abc^	2.35 ± 0.17 ^ab^	2.44 ± 0.31 ^a^	1.54 ± 0.14 ^c^
Leucine *	0.77 ± 0.12 ^b^	0.82 ± 0.05 ^ab^	0.97 ± 0.09 ^a^	0.90 ± 0.06 ^ab^	0.79 ± 0.03 ^b^
Lysine *	1.05 ± 0.21 ^b^	1.17 ± 0.12 ^ab^	1.35 ± 0.09 ^ab^	1.45 ± 0.21 ^a^	1.16 ± 0.08 ^ab^
Total EAAs	7.40 ± 1.29 ^a^	7.13 ± 0.70 ^a^	8.63 ± 0.55 ^a^	8.20 ± 0.87 ^a^	6.37 ± 0.10 ^b^
Total AAs	18.10 ± 3.19 ^ab^	16.89 ± 1.43 ^b^	20.76 ± 1.15 ^ab^	21.42 ± 2.42 ^a^	17.82 ± 0.42 ^ab^

Data are expressed as mean ± standard deviation (*n* = 3). Significant differences (*p* < 0.05) are indicated with different superscript letters in the same row. * Essential amino acids (EAAs).

**Table 3 foods-12-03311-t003:** Effects of different drying methods on the bioactive compounds of broccoli.

Parameter	Drying Method				
	VD	CD	IRD	LTVD	FD
TPC (mg GAE/g d.m.)	1.90 ± 0.11 ^b^	2.17 ± 0.22 ^a^	1.90 ± 0.26 ^b^	2.18 ± 0.19 ^a^	2.12 ± 0.22 ^a^
TFC (mg QE/g d.m.)	3.64 ± 0.40 ^c^	4.72 ± 0.30 ^b^	4.56 ± 0.49 ^b^	3.32 ± 0.36 ^c^	7.13 ± 0.71 ^a^
Chlorogenic acid (µg/g d.m.)	12.73 ± 0.73 ^d^	15.88 ± 0.27 ^c^	1.86 ± 0.11 ^e^	27.99 ± 0.18 ^a^	19.51 ± 2.44 ^b^
Caffeic acid (µg/g d.m.)	ND	ND	ND	4.60 ± 0.41 ^b^	6.76 ± 0.38 ^a^
Ferulic acid (µg/g d.m.)	5.19 ± 0.35 ^c^	11.45 ± 0.07 ^b^	3.09 ± 0.25 ^c^	36.93 ± 2.83 ^a^	11.35 ± 0.77 ^b^
TGC (µmol SE/g d.m.)	27.41 ± 3.33 ^c^	32.49 ± 2.22 ^b^	28.88 ± 3.84 ^bc^	30.17 ± 4.99 ^bc^	39.75 ± 4.20 ^a^
SFN (mg/g d.m.)	10.25 ± 1.67 ^b^	15.28 ± 3.04 ^a^	6.58 ± 1.93 ^c^	5.76 ± 0.73 ^c^	2.43 ± 0.13 ^d^

Data are expressed as mean ± standard deviation (*n* = 3). Significant differences (*p* < 0.05) are indicated with different superscript letters in the same row. Abbreviations: total phenolic content (TPC); total flavonoid content (TFC); total glucosinolate content (TGC); sulforaphane content (SFN); gallic acid equivalent (GAE); quercetin equivalent (QE); sinigrin equivalent (SE); and ND: not detected.

**Table 4 foods-12-03311-t004:** Activity of broccoli extracts on mouse ear edema inflammation brought on by phorbol-12-myristate-13-acetate (TPA) and arachidonic acid (AA).

Drying Method	Topical Anti-Inflammatory Effects		
	Dose (mg/ear)	%AA_AA_ ± SEM	%AA_TPA_ ± SEM
VD	3.0	26.9 ± 5.8	37.8 ± 14.2
CD	3.0	46.1 ± 5.4 *	56.1 ± 6.1 *
IRD	3.0	42.0 ± 8.6 *	39.8 ± 12.4 *
LTVD	3.0	55.4 ± 4.2 *	62.7 ± 5.4 *
FD	3.0	34.2 ± 3.9 *	43.2 ± 13.5 *
NIM	1.0	↑53.4 ± 8.2 *	n.d
IND	0.5	n.d	↑74.0 ± 13.8 *

AA_TPA_: topical anti-inflammatory activity against phorbol 12-myristate 13-acetate; AA_AA_: topical anti-inflammatory activity against arachidonic acid; NIM: nimesulide; and IND: indomethacin. n.d: not determined; ↑ maximal activity of the reference drugs. Significant differences (*p* < 0.05) between the samples in comparison to the negative control (100% inflammation) are indicated with an asterisk (*).

**Table 5 foods-12-03311-t005:** IC50 (mg/mL) values of extracts obtained from dried broccoli via different methods against MDA-MB-231 cells, calculated after 24 h of exposure using a best-fit regression model.

Drying Method	Equation (y = a ln(x) + b)		R^2^	* IC50 (mg/mL)
	a	b		
VD	39.51 ± 2.52 ^a^	57.42 ± 2.14 ^b^	0.975 ± 0.003 ^a^	0.830 ± 0.044 ^a^
CD	32.20 ± 0.75 ^b^	79.85 ± 0.88 ^a^	0.888 ± 0.005 ^b^	0.396 ± 0.019 ^b^
IRD	41.48 ± 1.11 ^a^	55.57 ± 1.36 ^b^	0.957 ± 0.000 ^a^	0.874 ± 0.032 ^a^
LTVD	34.13 ± 0.93 ^b^	78.55 ± 0.98 ^a^	0.888 ± 0.003 ^b^	0.433 ± 0.022 ^b^
FD	39.58 ± 1.35 ^a^	58.22 ± 7.76 ^b^	0.968 ± 0.016 ^a^	0.847 ± 0.176 ^a^

* The IC50 value was defined as the concentration of extracts that inhibited MDA-MB-231 cell growth in vitro by 50%. Data are expressed as mean ± SEM of three independent experiments (*n* = 3). Significant differences (*p* < 0.05) are indicated with different superscript letters in the same row.

## Data Availability

The datasets generated for this study are available on request to the corresponding author.

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
