# Peer review of "Low-Temperature Vacuum Drying on Broccoli: Enhanced Anti-Inflammatory and Anti-Proliferative Properties Regarding Other Drying Methods"

_foods, 2023, doi:10.3390/foods12173311_

Round 1

Reviewer 1 Report

The presented manuscript investigates the anti-inflammatory and anti-proliferative properties of low-temperature vacuum-dried broccoli. These properties were compared with the other four well-known drying methods. The subject is interesting, the manuscript is well-structured, and presents new and valuable results that be of great interest to the researchers in the field. 

However, in what follows, some remarks are outlined, which the authors might consider:

Page 2, line 96 – citation is needed for AOAC method.

It is not necessary to repeat the drying methods abbreviation under each table

From my point of view, some recently published papers should be cited and included in the discussion:

A.HAMEED et al., “Effect of different drying and cooking treatments on phytochemicals and antioxidant activity in broccoli: an experimental in vitro study”, Food Sci. Technol, Campinas, 2023, 43, e101622

X.MOHAMMADI et al. “Impact of three different dehydration methods on nutritional values and sensory quality of dried broccoli, oranges, and carrots”, Foods, 2020, 9, 1464

C. BAS-BELLVER et al., Impact of Disruption and Drying Conditions on Physicochemical, Functional and Antioxidant Properties of Powdered Ingredients Obtained from Brassica Vegetable By-Products. Foods, 2022, 11, 3663

P. BHATT et al. “Optimization of nutritional composition, bioactive compounds and antioxidant activity in Broccoli (Brassica oleracea) microgreen sprinkler using alternate drying techniques”, Asian Journal of Dairy and Food Research, 2023, 

Author Response

La Serena, August 29th, 2023

To Ms. Stella Song

Assistant Editor

Foods

Dear Ms. Song:

We appreciate you giving us the chance to revise our manuscript. As indicated, we have reduced the repetition rate in this version. Additionally, we responded to Reviewer#1 comments and have made some required edition. Changes in the manuscript have been highlighted. Please find below our itemized response to Reviewer#1 comments.

Kindly

Dr. Antonio Vega-Galvez

Food Engineering Department

University of La Serena

Eduardo Bitrán 1305

La Serena, Chile

Reviewer#1 comments

  1. The presented manuscript investigates the anti-inflammatory and anti-proliferative properties of low-temperature vacuum-dried broccoli. These properties were compared with the other four well-known drying methods. The subject is interesting, the manuscript is well-structured, and presents new and valuable results that be of great interest to the researchers in the field. However, in what follows, some remarks are outlined, which the authors might consider:

Authors: We are grateful for the helpful comments that reviewer #1 has made. Please find below our itemized responses to your comments.

  1. Page 2, line 96 – citation is needed for AOAC method.

Authors: We agree, it has been included the citation for AOAC method

  1. It is not necessary to repeat the drying methods abbreviation under each table

Authors: We have removed the drying methods abbreviation under each table according to your recommendation.

  1. From my point of view, some recently published papers should be cited and included in the discussion:
  • HAMEED et al., “Effect of different drying and cooking treatments on phytochemicals and antioxidant activity in broccoli: an experimental in vitro study”, Food Sci. Technol, Campinas, 2023, 43, e101622
  • MOHAMMADI et al. “Impact of three different dehydration methods on nutritional values and sensory quality of dried broccoli, oranges, and carrots”, Foods, 2020, 9, 1464
  • BAS-BELLVER et al., “Impact of Disruption and Drying Conditions on Physicochemical, Functional and Antioxidant Properties of Powdered Ingredients Obtained from Brassica Vegetable By-Products.” Foods, 2022, 11, 3663
  • BHATT et al. “Optimization of nutritional composition, bioactive compounds and antioxidant activity in Broccoli (Brassica oleracea) microgreen sprinkler using alternate drying techniques”, Asian Journal of Dairy and Food Research, 2023,

Authors: We have cited and included some recommended papers by Reviewer#1 in the discussion section. 

Reviewer 2 Report

I have carefully reviewed the manuscript titled Low temperature vacuum drying on broccoli (Brassica oleracea var. italica): Enhancement of its anti-inflammatory and anti-proliferative properties regarding other drying methods. First I would like to commend the authors for a very well done work. The manuscript is well-prepared and interesting. Some parts can be improved in terms of writing and clarity. Below are the comments which need to be addressed.

I suggest rephrasing the title. Adjust it since the phrasing should be more concise.

In the abstract section, I suggest improving clarity. Some sentences are quite lengthy, making them harder to read and understand. Breaking them down can improve clarity.

The introduction section is fine content-wise, but the writing must be improved.

If possible, I suggest preparing a schematic diagram of the process also it would be interesting to visually see the differences between dried materials. Was there any change in the color?

Line 178 – Method is missing reference.

Line 226 – How many mice were used in the experiment?

The results and discussion section are well done, but writing corrections are necessary.

Line 294 – Remove spaces between average values and deviation. This will improve the table.

In the conclusions section, I suggest adding future directions of the study as well as what were the limitations to your study and what can be built upon, etc.

Moderate editing of the English language is required.

Author Response

La Serena, August 29th, 2023

Ms. Stella Song

Assistant Editor

Foods

Dear Ms. Song:

We appreciate you giving us the chance to revise our manuscript. As indicated, we have reduced the repetition rate in this version. Additionally, we responded to Reviewer#2 comments and have made the required editions. Changes in the manuscript have been highlighted. Please find below our itemized responses to Reviewer#2 comments.

Kindly

Dr. Antonio Vega-Galvez

Food Engineering Department

University of La Serena

Eduardo Bitrán 1305

La Serena, Chile

Reviewer#2 comments

I have carefully reviewed the manuscript titled Low temperature vacuum drying on broccoli (Brassica oleracea var. italica): Enhancement of its anti-inflammatory and anti-proliferative properties regarding other drying methods. First I would like to commend the authors for a very well done work. The manuscript is well-prepared and interesting. Some parts can be improved in terms of writing and clarity. Below are the comments which need to be addressed.

Authors: We are grateful for the helpful comments that reviewer #2 has made. Please find below our itemized responses to your comments. Changes in the manuscript were highlighted in green.

  1. I suggest rephrasing the title. Adjust it since the phrasing should be more concise.

Authors: We agree, the title of manuscript has been shorted according to your recommendation.

  1. In the abstract section, I suggest improving clarity. Some sentences are quite lengthy, making them harder to read and understand. Breaking them down can improve clarity.

Authors: We have paraphrased and improved the clarity in the abstract section according to your recommendation.

  1. The introduction section is fine content-wise, but the writing must be improved.

Authors: We have improved the section according to the reviewer recommendation.

  1. If possible, I suggest preparing a schematic diagram of the process also it would be interesting to visually see the differences between dried materials. Was there any change in the color?

Authors: We consider it a valid suggestion. In fact, the schematic diagram was prepared as a graphical abstract requested by the journal. According to your suggestion, we have modified the graphical abstract and added images of the dried materials to see the different colors (see below) changes. We have also instrumentally measured the color, the results will be shown in our next work.

  1. Line 178 – Method is missing reference.

Authors: We have paraphrased and added the reference in the TFC method.

  1. Line 226 – How many mice were used in the experiment?

Authors: The statistical analysis of the experimental design was carried out by the biostatistician Dr. Luis Rodríguez from “Instituto de Salud Pública de Chile” (ISP). It aimed to detect differences between the treated groups and the negative control group considering that: i) each mouse, subtracting the weight of the right ear, minus the weight of the left ear of each animal and that ii) the edemas of the negative control group are compared independently of the treated group. It was determined that to calculate the percentage of anti-inflammatory effect, the median of the weights of the edemas of the negative control group and the median of the weights of the edemas of the group treated with the extracts under study should be used, establishing that the minimum size is of 8 mice per treated group and that for each treated mouse 2 control mice are required (16 controls).

The following table explains the number of animals used in the research:

Drying methods

Mice used in the evaluation of anti-inflammatory activity

n TPA

n AA

VD

8

8

CD

8

8

IRD

8

8

LTVD

8

8

FD

8

8

IND (TPA reference drug)

8

n.d

NIM (AA reference drug)

n.d

8

Control

16

16

TPA: phorbol 12-myristate 13-acetate; AA: arachidonic acid. NIM: nimesulide; IND: indomethacin. n.d: not determined,

In addition, we have added the number of mice used in the experiment in materials and methods section (control animals (n=16) and treated animals (n=8)).

  1. The results and discussion section are well done, but writing corrections are necessary.

Authors: We have improved the writing in the results and discussion section. Changes in the writing have been highlighted in yellow and green.

  1. Line 294 – Remove spaces between average values and deviation. This will improve the table.

Authors: We have improved the Table 1 according your recommendation.

  1. In the conclusions section, I suggest adding future directions of the study as well as what were the limitations to your study and what can be built upon, etc.

Authors: We have added future directions of the study and limitations of the study according your recommendation.

Comments on the Quality of English Language: Moderate editing of the English language is required.

Authors: It was done.
